# An Optimal Water Resource Allocation Mechanism Based on Ex-Post Verification and Reward in Huangbai River

## Hui Zhang [1] and Jiaying Li [2,*]

1   School of Economics and Management, Wuhan University, Wuhan 430072, China;
    hattie-zhanghui@whu.edu.cn
2   School of Economics and Management, Hubei University of Science and Technology, Xianning 437100, China
*   Correspondence: lijiaying@whu.edu.cn

**Abstract:** Under the current administrative system (AS) in China, the water resources governor allocates limited water resources to several users to realize the utility of water resources, leading to a principal–agent problem. The governor (referred to as the principal and she) wishes to maximize water resource allocation efficiency, while each user (referred to as the agent and he) only wishes to maximize his own quota. In addition, the governor cannot know water demand information exactly since it is the water users' private information. Hence, this paper builds an ex ante improved bankruptcy allocation rule and an ex post verification and reward mechanism to improve water allocation efficiency from the governor's perspective. In this mechanism, the governor allocates water among users based on an improved bankruptcy rule before the water is used up, verifies users' information by various approaches, and poses a negative reward to them if their information is found to be false after the water is used up. Then, this mechanism is applied to Huangbai River Basin. Research results show that the improved allocation rule could motivate users to report demand information more honestly, and ex post verification could motivate water users to further report their true information, which, as a result, could improve the water allocation efficiency. Furthermore, this mechanism could be applied to the allocation of other resources.

**Keywords:** water resource allocation; water allocation efficiency; ex ante improved bankruptcy allocation rule; ex post verification and reward; Huangbai River

## 1. Introduction

Water resources are one of the most basic and important elements for human beings. Global water demand has increased significantly over the past few decades due to population growth, climate change, and economic development, leading to a severe water crisis all over the world, especially in China, where water resources are unevenly distributed in spatial and temporal dimensions. However, the traditional water allocation method mainly takes the form of the administrative system (AS) [1], in which the water is distributed mainly according to the experience of the governor, resulting in inefficiency of water allocation and usage and further aggravating water resource shortage [2]. Therefore, it is crucial to employ other mechanisms under the AS to improve water resource allocation efficiency and to maintain the sustainable development of water resources.

Meanwhile, in China, the water resources governor has to allocate water resources to several users to realize water resource utility under the AS [3], giving rise to a principal–agent issue. This issue is about asymmetric information and conflicts of interests between the governor and water users, for which scholars have widely adopted the reward theory. For example, Kling [4] introduced economic incentive to improve water quality, Kahn et al. [5] employed political promotion incentive, φrum et al. [6] used economic transfer incentive. All of these showed that the reward schemes were very useful to deal with water resource management issues. However, in spite of lots of research results, there is still a huge potential to study the water allocation issue based on a mechanism design approach.

Hence, this paper employs an ex post verification and reward mechanism [7–10] to improve water resource efficiency under the current AS in China from the governor's view.

In the process of water resource allocation, the governor wishes to maximize water resource allocation efficiency, while each user only wishes to maximize his own quota. To make things worse, the governor cannot know water consumption information exactly, which is the water users' private information and would lead to moral hazard issues. This mechanism involves the interactions between the governor (referred to as she) and $n(n \geq 2)$ competing users (each of whom is referred to as he). The whole process includes two steps (the ex ante allocation rule and the ex post verification mechanism) and two key points (verification and reward). She announces this mechanism to each water user at the very beginning, and he chooses to accept it or not. If accepted, she faces all users with different demand information and requires them to report their information before allocation. Then she allocates water resources based on the bankruptcy rule with the reported information [11,12], which may induce users to exaggerate their values to a certain degree. After the water is used up, the governor will verify their information with probability and implement penalties if their information is found to be false. Here, the verification process means collecting various kinds of information such as production information and cost information and differentiating the true information from the false one. That is to say, the verified user who lies will be punished. Results show that the ex ante allocation rule based on improved bankruptcy theory could improve water resource allocation efficiency. Meanwhile, ex post verification and reward could motivate water users to report their true information, which could address the problem of information asymmetry and improve water resource allocation efficiency further. Therefore, this mechanism could help the governor decrease the degree of information asymmetry and improve water allocation efficiency. Additionally, it could also provide a new insight for water resource allocation management and relative decision-making support for the governor, which could be further applied to the allocation of other resources.

The main contributions of our paper are twofold, both in the theoretical aspect and in the practical aspect. Theoretically, from the governor's perspective, this paper sets up an ex post and reward mechanism to encourage all water users to act according to the governor's interests, which could maximize the efficiency of water resource allocation. Given water allocation inefficiency caused by asymmetric information, the governor uses two methods to motivate each user to report true information. First, she adopts the improved bankruptcy theory to allocate water resources, so that, instead of reporting water demand at will, the user's reported demand matches with his water resource contribution rate and GDP contribution rate. Second, the post-verification mechanism will reveal the true information of water demand, and those reporting false information will be punished, thus further ensuring the authenticity of information. Practically, a new mechanism is applied for water resource allocation in Huangbai River to explore the improvement of water resource allocation efficiency, and it could also be applied in other regions in China.

The paper is organized as follows: Section 2 presents a brief review of relevant literature, followed by some basic information about this mechanism in Section 3, such as event timeline and the allocation model. In Section 4, we present a case study in Huangbai River in China, including study area, the optimal mechanism, and water users' behaviors. The final section draws some conclusions.

## 2. Literature Review

The literature most related to our work was reviewed from the following four angles: water resource management, quota allocation, mechanism design with ex post verification and reward, and relative case studies.

In terms of water resource management, scholars introduced various models to deal with related issues [13,14], including multi-objective optimization model, programming method, improved bankruptcy rule, and so on. Li and Guo [15] proposed a multi-objective optimal water resource allocation model for irrigation water resources, which takes into

consideration economic benefits, social benefits, and ecological benefits to offer alternative irrigation water allocation plans under different scenarios for decision-makers. Bourque et al. [16] explored how to achieve a multibenefit optimization model of agriculture in terms of agricultural production, groundwater management, and biodiversity goals. Zhou et al. [17] employed a system dynamics–multiple objective optimization (SD-MOO) model based on "prediction + dynamic regulation + optimization" to study water resource management, which he applied in Jiaxing City and proved to be an effective tool to achieve sustainable water resource management. Mianabadi et al. [18] first introduced the water resource contribution rate into bankruptcy in water resource allocation, an important factor for improved bankruptcy rule. Li et al. [19] set up another improved bankruptcy model based on the water resource contribution rate, the efficiency of water-usage, and the minimum satisfied water demand and applied it in Dongjiang River. Furthermore, bankruptcy rule with game theory is another study direction. Janjua and Hassan [20,21] built stochastic bankruptcy games in transboundary water resource allocation based on GDP contribution and water resource utility. These multi-objective optimization issues are all decision-making problems involving one player but not multiparty negotiation and bargaining issues in current water resource management.

Quota allocation has been widely studied by many scholars in various areas, such as fishery, pollution emission rights, carbon emission quotas, and so on. Hatcher [22] studied an ITQ fishery with quota demands and the equilibrium quota price, which could be applicable to quota markets in pollution permit markets and carbon emission rights. Malik [6] researched self-report status and the environmental regulating mechanism based on the principal–agent issue, combining both monitoring and penalty tools, most related to our research. Results showed that self-reporting was desirable dependent on the frequency of audit and penalty. Bruno and Sexton [23] used market-based instruments relative to command and control to regulate groundwater trade, considering various buyers for the short-run trade market. Carbon emission quotas have been studied for a long time as well. Cui et al. (2021) [24] used a zero-sum gains data envelopment analysis (ZSG-DEA) model based on the entropy method to redistribute the carbon quotas across 30 provinces according to the principles of equality, efficiency, and sustainability. Huang and Xu [25] proposed a bi-level multi-objective model for carbon emission quota allocation under an uncertain environment, which could provide reasonable and practical strategies for the authority related to carbon emission quota allocation. As for water quota allocation, Wang [26] used the computational method of the quantity of eco-water requirement to balance water demand and maximal quota of water consumption, a case study in Chaoyang Pak. Moreover, quota allocation suits water resource allocation mechanism under the AS, even though there exists asymmetric information and conflicts of interest. This is exactly the main motivation of this paper.

In terms of mechanism design based on verification and punishment, Townsend [27] first proposed verification on the principal–agent model with optimal contract, and they believed verification is deterministic. Gale and Hellwig [28] introduced verification cost in the credit market and Ben-Porath et al. [8] employed costly verification of the allocation mechanism. These models differ from what we consider, in that, they only focus on verification but not punishment. Mylovanov and Zapechelnyuk [9] also introduced a mechanism with free ex post verification and limited penalty. Similarly, Li [10] designed a mechanism with costly verification and limited punishment. Li [29] and Patel and Urgun [30] also made contributions in these issues. According to their research, the allocation mechanism with verification and punishment could provide an important perspective for the issue of allocating an indivisible object only to one agent, while divisible resource allocation, such as the agricultural water resource allocation, has not been studied. This is the core motivation of this paper. The cross-subsidy strategy [11,12] has also been used in water resource management. On the one hand, cross-subsidy strategy could give an incentive to all participants. On the other hand, this method has not been coupled with ex post verification to address water resource allocation issues.

In addition, case studies related to the water allocation mechanism have been widely conducted all over the world. Wang et al. [31] introduced a case study of the Heihe River Basin to examine agricultural water use efficiency (WUE) based on three-stage DEA (data envelopment analysis). Rolfe and Windle [32] used an auction mechanism to address water quality improvement issues in Great Barrier Reef catchments by revealing opportunity costs in Australia, which could be used to design a better allocation mechanism and set reserve prices for future water quality tenders. Yao et al. [33] proposed a multi-objective multistage Stackelberg–Nash–Cournot game model to cope with conflicts concerning the optimum allocation of water resources under various climate scenarios between regional authorities and subarea managers and applied this model in the Dujiangyan Irrigation System in China to identify the tradeoffs. Xu et al. [34] employed a multi-objective water resource allocation model based on equity and efficiency principles and applied it in the Minjiang River Basin, demonstrating the practicality and rationality of the proposed mechanism. Similar mechanisms also apply to the Murray–Darling Basin in Australia [35], Iran [36], and so on. Compared with previous research, these proposed mechanisms provide new insights related to water resource allocation. Relative case studies also prove the feasibility and adaptability of proposed mechanisms. Hence, in this paper, we also build a water resource allocation mechanism and apply it in the Huangbai River.

Our work extends the ex post verification and reward mechanism to water resource allocation management with multiple competing users. It sets up a water resource allocation mechanism including ex ante improved bankruptcy allocation rule, ex post verification, and reward to improve the efficiency of water resource allocation and usage with a case study. This mechanism aims to motivate all competing users to act in line with the governor's interests rather than at the expense of the governor, which could be used for the allocation of other resources.

## 3. The Model

Our model best suits those governors who adopt principal–agent theory to improve water resource allocation efficiency under the AS. In this model, we emphasize the interaction between the governor and $n(n \geq 2)$ competing water users.

In this section, we provide an event timetable first; then we put forward the basic information about this model, including self-report information, allocation rule, and ex post verification and reward mechanism. Note that the allocation rule before the water is used up and the ex post verification and penalty after the water is used up are two key points in our analysis: (1) On the one hand, before the water is used up, the governor allocates water resources based on the improved bankruptcy rule in a relatively fair and efficient way; on the other, the improved bankruptcy rule plays the role of cross-subsidy, rewarding each user positively or negatively, encouraging them to report demand information related to their own water resource contribution and GDP contribution. (ii) After the water is used up, ex post information verification reveals users' true information, and the governor poses a penalty to user *i* if his information is found to be false, hence motivating each user to report true information. This will further push all competing water users to act on the governor's behalf and help the governor realize her original goals.

### 3.1. Event Timeline

According to the previous analysis, the timeline of the mechanism is as follows, from the governor's perspective (Figure 1) and from user *i*'s perspective (Figure 2).

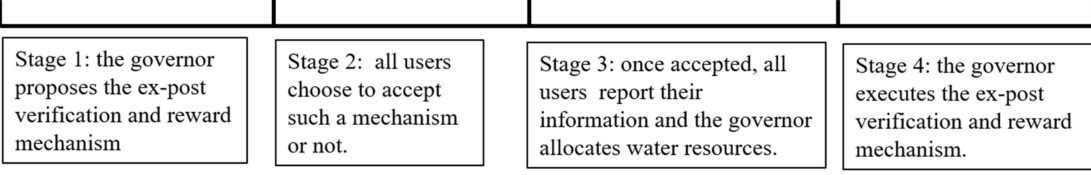

**Figure 1.** Timeline of the mechanism from the governor's perspective.

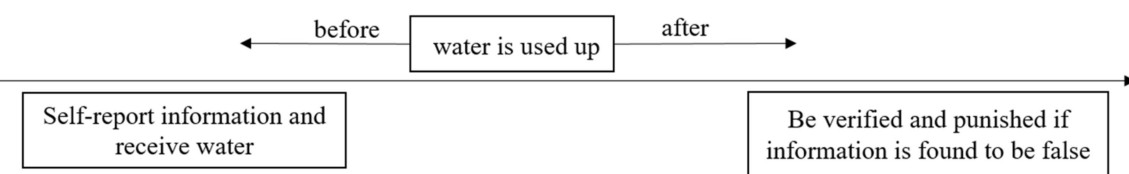

**Figure 2.** Timeline of the mechanism from user $i$'s perspective.

### 3.1.1. Timeline of the Mechanism from the Governor's Perspective

From the governor's perspective, there are four steps of the event timeline.

First, the governor proposes the predetermined ex post verification and reward to all competing water users about their demand information before allocation; the governor has full commitment to this mechanism.

Second, each user chooses to accept such a mechanism or not.

Third, once accepted, each user has to report his demand and the governor allocates water resources according to his reported information based on bankruptcy rule.

Finally, the governor executes the ex post verification and reward mechanism.

### 3.1.2. Timeline of the Mechanism from the Governor's Perspective

From user $i$'s perspective, once he accepts this mechanism, there are two key stages in this mechanism: before the water is used up and afterwards.

First, before the water is used up, user $i$ has to report demand information and receive water resources based on the improved bankruptcy rule.

Second, after the water is used up, the governor inspects water demand information and poses a penalty if information is found to be false.

### *3.2. Basic Information about the Model*

Based on Figures 1 and 2, the basic information of this model contains information about following aspects: the predetermined mechanism, self-report information, water resource allocation, ex post verification, and reward.

### 3.2.1. Predetermined Mechanism Analysis

The governor proposes the predetermined mechanism, including the allocation rule, $\{t_i\}$, and the ex post verification and reward mechanism, $\{p_i, f_i\}$. The following part gives the details about self-report information and $\{t_i\}$ and $\{p_i, f_i\}$.

### 3.2.2. Self-Report Information Analysis

Suppose that the set of users is $N := \{1, \ldots, n\}$, which means $n$ different users in the same area. Each user has to submit his demand information, $u_i(d_i, \omega_i)$, to the governor, which is decided by his true private demand information $d_i$ and user $i$'s control effort level $\omega_i$ related to self-reporting [37]. The cost related to $u_i(d_i, \omega_i)$ is $C_i(\omega_i)$, $C(0) = 0$, $C'(\omega_i) > 0$, $C''(\omega_i) > 0$. Moreover, $u_i \in [u, \overline{u}]$, where $u$ is the lowest demand quantity, and $\overline{u}$ is the highest demand quantity according to the history of the governor's allocation. As a rational user, $u_i \geq d_i$.

### 3.2.3. Quota-Based Allocation Rule Analysis

The governor allocates water resources to users based on the improved bankruptcy rule related to the reported demand information. The following is the improved bankruptcy rule based on the basic and classical bankruptcy rule.

(i) Basic bankruptcy rule

$$\begin{cases} X = \sum_{i \in N} x_i \\ X \leq \sum_{i \in N} u_i \\ 0 \leq x_i \leq u_i \end{cases} \tag{1}$$

where $X$ is the total available water resources, $\sum_{i \in N} u_i$ is the total demand quantity, $x_i$ is the amount of water resources assigned to user $i$.

(ii) Classical bankruptcy rule

There are three classical bankruptcy rules: P rule, CEA rule, and CEL rule.

(1) P rule

$$x_i^p = \alpha u_i, \text{ where } \alpha = \frac{X}{\sum_{i=1}^{N} u_i} \tag{2}$$

(2) CEA rule

$$x_i^{CEA} = \min\{\beta, u_i\}, \text{ where} \beta = \frac{X}{n} u_m < \beta(m \in \{1, \ldots, n\}), x_m = u_m, \tag{3}$$

$x_i^{CEA}(i \neq m) = \min\{\beta^{\%_0}, u_i\}, \text{ where } \beta' = \frac{(X - u_m)}{(n-1)}$

(3) CEL rule

$$x_i^{CEL} = \max\{0, u_i - \chi\}, \text{ where} \chi = \frac{(\sum_{i=1}^{N} u_i - X)}{n} \tag{4}$$

When $x_k < \chi, k \in \{1, \ldots, n\}, x_k = 0,$,

$x_i^{CEL}(i \neq k) = \max\{0, u_i - \chi'\}, \text{ where } \chi' = \frac{(\sum_{i=1}^{N} u_i - X)}{(n-1)}$

(iii) Improved bankruptcy rule

The improved bankruptcy rule considers two factors with water resources: the first factor is the contribution of regional water resources [18], $a_i$, and the second is the contribution of regional GDP [19], $I_i$, which could be expressed as follows.

(1) When $\sum_{i=1}^{N} u_i > X$, supply falls short of demand.

$$x_i = u_i - t_i \tag{5}$$

$$t_i = \frac{(\frac{u_i}{\sum_{i \in N} u_i} + 1 - \eta_i + e_i)}{n+1} * (\sum_{i=1}^{N} u_i - X) \tag{6}$$

$$\sum_{i \in N} a_i = \sum_{i \in N} x_i = X \tag{7}$$

$$0 \leq x_i \leq u_i \tag{8}$$

$$\eta_i = \frac{a_i}{\sum_{i \in N} a_i} \tag{9}$$

$$e_i = \frac{I_i}{\sum_{i \in N} I_i} \tag{10}$$

$$\sum_{i \in N} t_i = \sum_{i=1}^{N} u_i - X \tag{11}$$

According to Equations (5)–(11), in the case of water shortage, for user $i$, the more the water resource contribution and the less the GDP contribution he makes, the more water resources he will obtain. One reason is the water resource dispatching cost, and the other is preparing for the development of water market.

(2) When $\sum\limits_{i=1}^{N} u_i \leq X$, there is sufficient supply for all users.

To improve water resource allocation efficiency, the governor will allocate water resources based on improved bankruptcy to motivate each user to report true information related to his water resource contribution and GDP contribution.

$$x_i = \frac{\left(\frac{u_i}{\sum\limits_{i\in N} u_i} + 1 - \eta_i + e_i\right)}{n+1} \times \sum_{i=1}^{N} u_i \tag{12}$$

$$t_i = x_i - u_i \tag{13}$$

$$\sum_{i\in N} t_i = 0 \tag{14}$$

According to Equations (12)–(14), when supply and demand are in balance, the improved bankruptcy rule could motivate users to report water resources while the governor considers more factors to enhance current water resource allocation efficiency and to develop the water market for the long term. Then $\{t_i\}$ could be the cross-subsidy for all competing users. Additionally, for the governor, the cost related to the penalty is $c_{it}$.

### 3.2.4. Ex Post Verification and Reward Mechanism Analysis

After water resources are allocated and used up, the governor will execute an ex post verification and reward mechanism $\{p_i, f_i\}$ by setting the verification probability, $p_i$, and reward level, $f_i$, to urge all users to report information truthfully in future.

To verify users' reports, the governor could inspect each user with probability, $p_i$, if $u_i \in [u, \overline{u}]$ through all means, for which the cost of the governor is $C$. The governor will inspect users after the water is used up for agriculture or industry or other aims.

The other two situations are then considered in the following, and we assume $u_i \in [u, \overline{u}]$ in this paper.

$$x_i = \begin{cases} u_i, u_i < u, \\ x_i, u_i \in [u, \overline{u}] \\ u_i, u_i > \overline{u}, \end{cases}, p_i = \begin{cases} 0, u_i < u \\ p_i, u_i \in [u, \overline{u}] \\ 1, u_i > \overline{u} \end{cases}$$

Considering the difficulty of information acquisition, imperfect verification also exists. Let $\alpha_{ij}$ represent the probability that the governor infers the demand information, where $i \neq j$ means inferring incorrectly, and $i = j$ means inferring correctly. If the governor could infer users' information perfectly, $\alpha_{ii} = 1$, $\alpha_{ij} = 0$ $(i \neq j)$. Considering the costly verification, the correct inferring probability is larger than the incorrect one, $\alpha_{ii} > \alpha_{ij}$ $(i \neq j)$, and the $\alpha_{ij}$ is the knowledge shared by the governor and users.

The governor will impose the penalty $f_i$ for users if they were found to report false information, and the cost related to the penalty is $c_{iv}$.

### 3.3. The Objective Function of the Mechanism

Based on the previous analysis and the revelation principle, it is easy to see that the truthful reporting of users' information constitutes a Bayesian Nash equilibrium. Clearly, if user $i$ reports his information, which is verified afterwards, it is best to punish him if and only if he is found to lie and not to punish him otherwise.

Under this condition, a direct mechanism consists of an allocation rule, a verification rule, and a punishment rule, written as $\{p_i, f_i\}$,

Then the expected quantity of user $i$ who reports $u$ is $p_i(\alpha_{ii}(d_i - f_i) + \alpha_{ij}(u_i - f_i) - t_i^* - C(\omega_i)) + (1 - p_i)(u_i - t_i^* - C(\omega_i))$. The utility function of the user $i$ can be written as follows:

$$\text{Max } \left( p_i(\alpha_{ii}(d_i - f_i) + \alpha_{ij}(u_i - f_i) - t_i^* - C(\omega_i)) + (1 - p_i)(u_i - t_i^* - C(\omega_i)) \right) \tag{15}$$

For a risk-neutral user $i$, this mechanism is feasible if this $\{p_i, t_i, f_i\}$ satisfies $p_i \geq 0$ and satisfies the Bayesian incentive compatibility (BIC) constraints,

$$p_i(\alpha_{ii}(d_i - f_i) + \alpha_{ij}(u_i - f_i) - t_i^* - C(\omega_i)) + (1 - p_i)(u_i - t_i^* - C(\omega_i)) \geq u_0 \text{ (IC)} \tag{16}$$

where $u_0$ is user $i$'s reservation demand quantity.

The governor's objective is to maximize his expected utility by allocating the limited water resources in spite of the expected verification cost. Then the governor's objective function is

$$\max E\left[ \sum_{i \in N} (x_i - p_i C + t_i + p_i \alpha_{ij} f_i - c_{it} - p_i c_{iv}) \right] \tag{17}$$

The optimal mechanism is to set the optimal $\{p_i, f_i\}$ to induce the user to act in line with the governor's interests rather than his own. There are two steps to seek the optimal $\{p_i^*, f_i^*\}$ in water resource allocation.

First, set the optimal $t_i^*$, positive or negative, which could be referred to in Equations (5)–(14).

Second, set the optimal $\{p_i^*, f_i^*\}$ to realize the governor's initial aims. There are two cases related to the optimal mechanism, i.e., the first best mechanism and the second best mechanism.

(i) For the first best mechanism benchmark, there is no asymmetric information between the governor and users, which means each user reports true information, and the governor allocates water resources based on quotas. Then the optimal mechanism is $\{0, t_i^*, 0\}$.

(ii) In the second best mechanism, asymmetric information exists.

For the governor,

$$E\left[ \sum_{i \in N} (x_i - p_i C + t_i^* + p_i \alpha_{ij} f_i - c_{it} - p_i c_{iv}) \right] = E\left[ \sum_{i \in N} (x_i - p_i(C + c_{iv}) + u_i - u_0 - C(\omega_i) - c_{it}) \right] \tag{18}$$

$$p_i^* = \text{argmax} E\left[ \sum_{i \in N} (x_i - p_i(C + c_{iv}) + u_i - u_0 - C(\omega_i) - c_{it}) \right] \tag{19}$$

For user $i$, IC is binding,

$$p_i(\alpha_{ii}(d_i - f_i) + \alpha_{ij}(u_i - f_i) - t_i^* - C(\omega_i)) + (1 - p_i)(u_i - t_i^* - C(\omega_i)) = u_0 \tag{20}$$

$$p_i \alpha_{ii}(u_i - d_i) + p_i f_i = u_i - u_0 - t_i^* - C(\omega_i) \tag{21}$$

Then the optimal solution $\{p_i^*, f_i^*\}$ could be achieved based on Equations (14)–(16). It implies that all users will report information $u_i$, where the governor's original goals will be realized, and all users will act according to the governor's interests rather than at her expense.

**Theorem 1.** *If $\{p_i^*, t_i^*, f_i^*\}$ is the optimal solution from the governor's perspective, it implies that all users will report information $u_i$, where the governor's original goals will be realized and all users will act according to the governor's interests rather than at her expense.*

**Proof.** As proved by the definition, if a player accepts this mechanism, an optimal mechanism satisfies the IC and IR constraint first. Meanwhile, if a mechanism is optimal, from the governor's perspective, it implies that all users will report information $u_i$, where the governor's original goals will be realized, and all users will act according to the governor's interests rather than at her expense.

Moreover, without considering the verification cost, $v_{iv}$, and cross-subsidy cost, $v_{it}$, the value of $p_i f_i$ will be certain, which means that if the verification frequency is high, the penalty level will be low, and vice versa. There is another situation considering the verification cost, $v_{iv}$, and cross-subsidy cost, $v_{it}$.

In the following part, we will discuss the optimal mechanism, $\{p_i^*, f_i^*\}$, through a case study, with and without all costs.

## 4. Case Study

In this paper, to reduce the water resource allocation inefficiency caused by asymmetric information between the governor and users under the current AS in China, we employ the principal–agent theory with ex post verification and reward for Huangbai River. Both analytical and numerical results are presented in this section.

For each town, the information regarding demand quantity is the town's private information when the governor requires it to report its demand quantity in order to allocate water resources. The governor's aim is to minimize the social cost and improve water resource efficiency, while each town only wishes to receive more water resources.

### 4.1. Study Area

Huangbai River is located in the north bank of the lower reaches of the Three Gorges of the Yangtze River in China. The main stream of the Huangbai River Basin is divided into two branches, the east and the west (as shown in Figure 3). The east branch is the research object of this paper, including two counties, i.e., Yiling District and Yuan'an County, and six towns, i.e., Zhangcunping Town, Wuduhe Town, Fenxiang Town, Huanghua Town, Xiaoxita Jieban Town in Yiling District, and Luozu Town in Yuan'an County. The basic information of these towns is listed in Table 1.

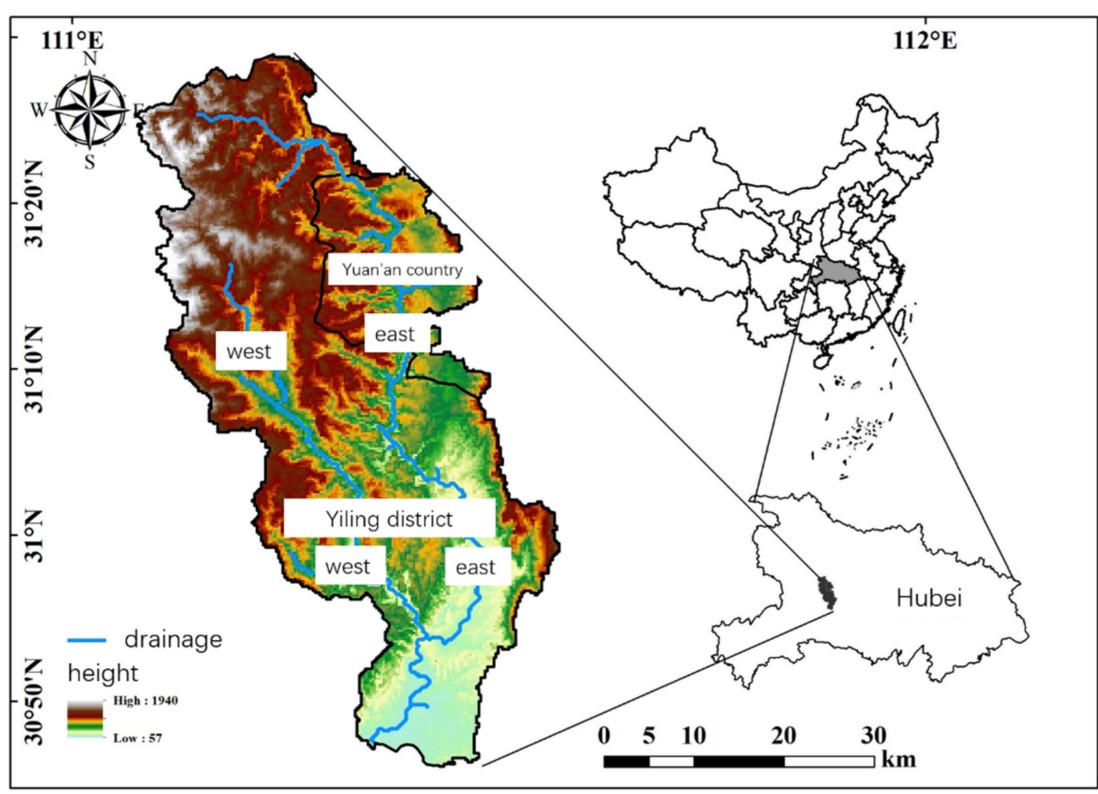

**Figure 3.** The drainage map of Huangbai River Basin.

**Table 1.** Basic information of Huangbai River.

| District | Town | Different Hydrological Conditions | Reported Water Demand, $u_i$ (Unit: Million m$^3$) | Water Resource Contribution, $a_i$ (Unit: Mil-lion m$^3$) | GDP Contribution, $I_i$ (Unit: Trillion CNY) |
|---|---|---|---|---|---|
| Yiling district | Zhangcunping | annual average | 615 | 86 | 65.2 |
| | | 95% | 588 | | |
| | Wuduhe | annual average | 1036 | 153 | 85.8 |
| | | 95% | 954 | | |
| | Fenxiang | annual average | 1128 | 33,040 | 104.9 |
| | | 95% | 1038 | | |
| | Haunghua | annual average | 1009 | 951 | 96.2 |
| | | 95% | 1019 | | |
| | Xiaoxita Jieban | annual average | 1789 | 16,619 | 332.5 |
| | | 95% | 1722 | | |
| Yuan'an | Luozu | annual average | 1295 | 5875 | 67.6 |
| | | 95% | 1346 | | |

### 4.2. The Optimal Mechanism Analysis

First, each town reported its own demand quantity before the governor allocated the water resources.

This paper analyzes water resources in this basin in 2025 under the hydrological conditions of annual average and 95% frequency. With the annual average, supply is equal to demand, with demand being 68.71 million m$^3$ and supply being 68.71 million m$^3$. With 95% frequency, there is water resource shortage. That is to say, supply is less than demand, and the allocated water demand and water resource supply are 66.67 and 58.94 million m$^3$, respectively, in the special dry year frequency, with a shortage of 7.73 million m$^3$.

In the first case, if there is no asymmetric information between the governor and users, the allocation results are as follows in Table 2:

**Table 2.** Water resource allocation with symmetric information.

| District | Town | Different Hydrological Conditions | Water Allocation, $X_i$ (Unit: Million m$^3$) | $t_i^*$ (Unit: Million m$^3$) |
|---|---|---|---|---|
| Yiling district | Zhangcunping | annual average | 615 | 0 |
| | | 95% | 519.83 | 68.17 |
| | Wuduhe | annual average | 1036 | 0 |
| | | 95% | 843.39 | 110.61 |
| | Fenxiang | annual average | 1128 | 0 |
| | | 95% | 917.65 | 120.35 |
| | Haunghua | annual average | 1009 | 0 |
| | | 95% | 900.85 | 118.15 |
| | Xiaoxita Jieban | annual average | 1789 | 0 |
| | | 95% | 1522.34 | 199.66 |
| Yuan'an | Luozu | annual average | 1295 | 0 |
| | | 95% | 1189.94 | 150.06 |

In the second case, there exists asymmetric information and user *i* reported his demand information based on his true and controlled effort level.

The first step is to set the optimal $t_i^*$ in Table 3.

The optimal cross-subsidy, $t_i^*$, considers three important factors, including reported demand quantity, water contribution, and GDP contribution. Compared with the first case, it could improve the allocation efficiency for these regions.

Second, set the optimal, $\{p_i^*, f_i^*\}$.

IC is binding, $p_i \alpha_{ii}(u_i - d_i) + p_i f_i = u_i - u_0 - t_i^* - C(\omega_i)$

For the governor, $p_i^* = \arg\max E[\sum_{i \in N}(x_i - p_i(C + c_{iv}) + u_i - u_0 - C(\omega_i) - c_{it})]$.

Take Zhangcunping Town for example. Considering the lowest satisfied water demand, $\alpha_{11} = 0.85$, we assume the control effort level $C(\omega_i) = a\omega_i^2 + b\omega_i$, $a = 1$, $b = 2$, and we could gain the following results without considering all costs (Figure 4).

**Table 3.** Water resource allocation with $t_i^*$ with symmetric information.

| District | Town | Frequency | Water Allocation (Unit: Million m³) | $t_i^*$(Unit: Million m³) |
|---|---|---|---|---|
| Yiling district | Zhangcunping | annual average | 985.8223 | 370.8223 |
| | | 95% | 844.5535 | 324.7287 |
| | Wuduhe | annual average | 1020.234 | −15.7659 |
| | | 95% | 868.7122 | 25.32297 |
| | Fenxiang | annual average | 1577.539 | 449.5387 |
| | | 95% | 1346.109 | 428.459 |
| | Haunghua | annual average | 1016.615 | 7.615078 |
| | | 95% | 877.1251 | −23.7278 |
| | Xiaoxita Jieban | annual average | 1090.796 | −698.204 |
| | | 95% | 933.972 | −588.372 |
| Yuan'an | Luozu | annual average | 1179.994 | −115.006 |
| | | 95% | 1023.528 | −166.411 |

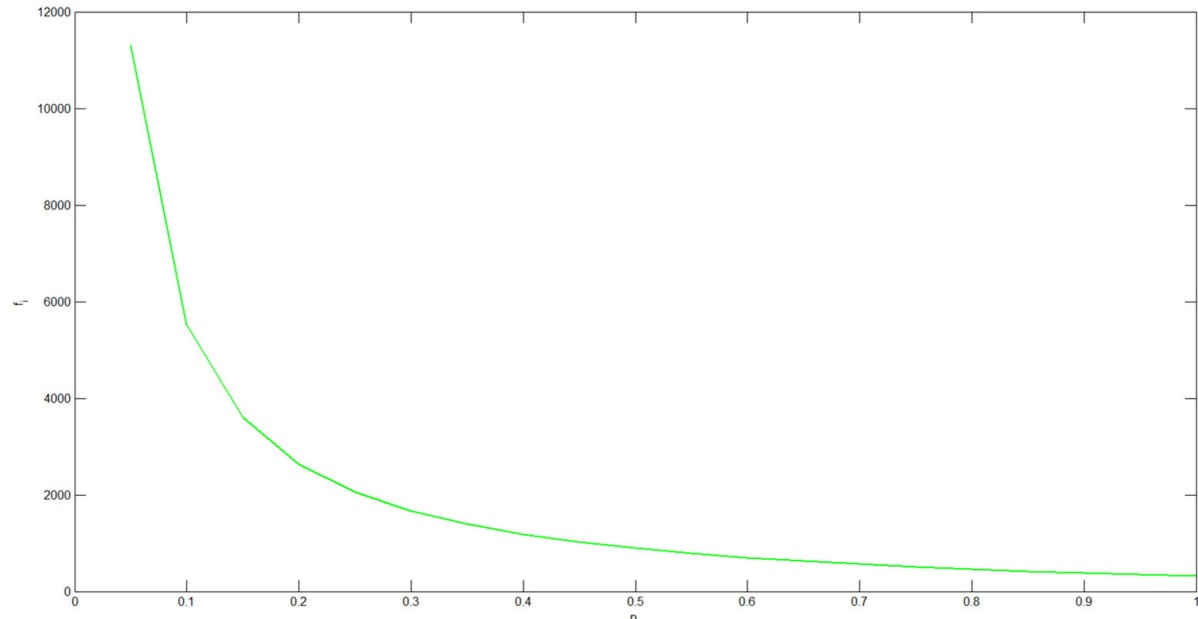

**Figure 4.** The optimal $(p_i^*, f_i^*)$ when $\alpha_{ii} = 0.85$.

According to these results, the optimal $(p_i^*, f_i^*)$ is shown in Figure 3, which could reflect the relationship between $p_i^*$ and $f_i^*$. The lower the verification frequency is, the higher the reward level is, and the lower $\alpha_{ii}$ is, the higher the reward level is. Moreover, when verification frequency, $p_i$, is low, the governor has to set a high penalty, $f_i$, to motivate users to report relatively true demand information [38].

Additionally, from user $i$'s perspective, if the reward exceeds his benefit from reporting false information, he will also choose to report true information, or he will choose to exert more efforts to use water resources more efficiently, which is worth studying in future research.

Based on the above analysis, there are two cases with and without ex post verification and reward. Compared with the first case, which involves symmetric information and no ex post verification and reward mechanism, the second case with the ex post verification

and reward mechanism could improve allocation efficiency for several reasons: (i) the optimal cross-subsidy, $t_i^*$, considering reported demand quantity, water contribution, and GDP contribution, could reflect the practice situations rather than just the reported values; (ii) the verification and reward, $(p_i^*, f_i^*)$, further motivate each user to report relatively true values and act according to the governor's interests rather than at the expense of the governor.

## 5. Conclusions and Discussion

### 5.1. Conclusions

This paper explores the improvement of water resource allocation efficiency by taking into account the asymmetric information and conflicts of interest between the governor and several users. A model based on ex ante improved bankruptcy allocation rule, ex post verification, and reward mechanism is proposed and applied in Huangbai River. There are two novel contributions in this paper, theoretically and practically.

From the theoretical aspect, (i) water resource allocation involves asymmetric information and conflicts of interest between the governor and water resource users, which can be dealt with through the principal–agent theory, including ex post verification and reward mechanism. (ii) The predetermined mechanism covers the whole process, including ex ante allocation and ex post verification that motivate each user to report relatively true demand information. An ex ante improved bankruptcy allocation rule could give users incentive to report demand based on his contribution, water resource contribution, and GDP contribution. Ex post verification and reward further give rise to true demand information. (iii) The existing improved bankruptcy rule in water resource allocation studies do not fully consider GDP contribution, which is crucial for the water market.

From the practical aspect, (i) this mechanism could help the governor improve water resource allocation efficiency from ex ante allocation and ex post verification under the current AS; (ii) ex post verification and reward have not been applied to divisible resources' allocation, which is an important research topic in future, such as water resource allocation and other resource allocation. (ii) This mechanism also has great guiding significance for the construction of a water right market and water resource allocation in Huangbai River and other regions in China, where asymmetric information exists, too.

### 5.2. Disussion

It is necessary to explore all possible solutions to address the water crisis because water resource allocation is a long-term project all over the world. In this paper, the ex post verification and reward mechanism could address the principal–agent issue related to water resource allocation, considering the asymmetric information and conflicts of interest between the governor and users. In fact, as users may come from different industries, such as agriculture, manufacturing, and others, ex ante improved bankruptcy allocation and ex post verification should be considered differently; users may cooperate with each other to report demand information, honestly or dishonestly, which will affect the allocation rules directly. Furthermore, verification cost maybe type-dependent, this is to say, each user may have a different verification cost, which also affects the governor's utility function.

Based on the above analysis, there are two main extension aspects for future investigation. First, we can extend different competing water users to several different types of users (more than two), including users with low demand, users with high demand, industrial users, and environmental users. The governor and various users will behave differently when the governor faces more choices and each user faces more competition. According to several previous papers [7–10], various types of users play an important role in mechanism design. The governor will face different utility functions related to water demand quantities, which will directly influence the total verification cost and relative reward and force all participants to change their behaviors. This is another interesting direction for future research. The second extension is that competing water users choose to cooperate with each other to bargain with the governor when facing the predetermined

allocation mechanism. That is to say, if users from different sectors or the same sector cooperate to bargain with the governor, they will have more bargaining power than one user alone, which will necessitate a different allocation mechanism, another interesting research direction left for further research.

It is also worthwhile to study those related issues from such directions as type-dependent costly verification for each user, free verification and very large punishment, repeated allocation management during a period of time, and so on.

**Author Contributions:** Conceptualization, H.Z. and J.L.; formal analysis, H.Z. and J.L.; funding acquisition, J.L.; investigation, H.Z.; methodology, H.Z.; resources, J.L.; software, J.L.; supervision, H.Z. and J.L.; validation, H.Z.; writing—original draft, H.Z.; writing—review and editing, H.Z. and J.L. All authors have read and agreed to the published version of the manuscript.

**Funding:** This research was funded by Hubei Social Science Foundation, grant number 2020187.

**Institutional Review Board Statement:** Not applicable.

**Informed Consent Statement:** Not applicable.

**Data Availability Statement:** http://slt.hubei.gov.cn/bsfw/cxfw/szygb/, http://www.stats.gov.cn/tjsj./ndsj/ (accessed on 3 June 2021).

**Acknowledgments:** This research was financially supported by the Hubei Social Science Foundation (2020187). The authors would like to thank the funded project for providing material for this research. We would also like to thank the editor and reviewers very much for their valuable comments in developing this article.

**Conflicts of Interest:** The authors declare no conflict of interest.

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
