# Peer review of "An Optimal Water Resource Allocation Mechanism Based on Ex-Post Verification and Reward in Huangbai River"

_water, doi:10.3390/w13111588_

Round 1
Reviewer 1 Report
see file

Author Response
We gratefully acknowledge the editor and the anonymous reviewers for the valuable suggestions and comments. After seriously considering the suggestions and comments, we prepared a revised version of the manuscript. The corresponding responses are listed as follows.

Reviewer 2 Report
The study excessively pivots around the key role of the governor as the main argument employed to provide underpinnings of the research. Moreover, literature review enumerates a set of contributions associated to water resources management, quotas allocation, mechanisms design and ex-post verification and reward, but no gaps were disclosed to be bridged by the manuscript. As a result, background is poor, whilst the objective of the study is unclear. It seems that a tailored mechanism was built to merely satisfy the wishes of the governor and its application to other geographical areas outside China is unlikely. Contributions depicted in lines 78-83 are ambiguous and irrelevant.
Hence, the first two sections should be focused on i) determine reasons why an ex-post verification is more suitable than an ex-ante one, ii) contextualize examined researches on ex-post verification to the Chinese case by properly introducing then governor´s requirements.
A general overview of the methodology is lacking to enable replicability by other authors. With regard to the prospective replicability, only strictly mathematical expressions should be displayed. Since Section 3 is highly confusing, a figure is recommended to connect methods of subsections 3.2 and 3.3. Furthermore, a specified research aim could help to understand if objective function defined in 3.3 is convenient or not.
Criteria to select Huangbai River as a case study are unknown. Discussion must be based on findings rather than future studies. No significant conclusions were provided.
Miscellaneous comments. English grammar and style should be enhanced in some sections. The manuscript must be organized in accordance to the sections defined by the journal (https://www.mdpi.com/journal/water/instructions): Introduction, Materials & Methods, Results, Discussion and Conclusions (not mandatory). Check format of references #26 and 27. The format of tables should be improved.
Author Response

(The authors gave the same response as above.)

Round 2
Reviewer 1 Report
All my comments have been taken into account in this new version. The letter with the reply was instructive concerning the possible confusion between variables. The paper is more readable now.
Line 323 page 8 and line 336 page 9 the formula x_i as a fonction of the u seems to appear twice. There is still a problem with writing the formulas in the text
Author Response
We gratefully acknowledge the editor and the anonymous reviewers for the valuable suggestions and comments. After seriously considering the suggestions and comments, we prepared a revised version of the manuscript.

Reviewer 2 Report
After reading the revision of the manuscript “An optimal water resources allocation mechanism based on ex-post verification and reward in Huangbai River”, I highlight next key remarks still unaddressed:
The study primarily pivots around the governor as the main argument employed to underpin the research. Despite the term “bankruptcy allocation rule” was now used in Abstract and Introduction, it was not even characterized. Literature review is mostly descriptive, without disclosing gaps to be addressed in the study. For instance, “bankruptcy allocation rule” was fully omitted. The research aim is ambiguous, beyond the fulfillment of the wishes of the governor.
A general overview of the methodology is lacking to enable replicability by other authors. Lines 197-214 are confusing. Only mathematical expressions strictly necessary should be displayed to enable prospective replicability.
Neither discussion of findings nor a summary of main conclusions were provided. Indeed, it is unclear how “this mechanism could help the governor alleviate the problem of distribution efficiency….” (lines 536-540)
Other comments. The length of Abstract cannot exceed a maximum of 200 words. It is unknown why citations appear in the manuscript as superscript numbers. Check citation in line 49. English grammar and style should be enhanced.
Author Response

(The authors gave the same response as above.)

Round 3
Reviewer 2 Report
..
This manuscript is a resubmission of an earlier submission. The following is a list of the peer review reports and author responses from that submission.
Round 1
Reviewer 1 Report
see file

Reviewer 2 Report
After reading the manuscript “An optimal agricultural water resources allocation mechanism based on ex-post verification and reward”, I highlight next remarks:
- Because diverse references to “ the governor” were found, the study was mostly characterized to the Chinese context. The title of the paper should reflect this point. However, it is very unclear its eventual application to other geographical areas.
- As stated by authors, water allocation is a topic widely examined by literature, mostly based on the study of previous demand/ real consumption. Underpinnings of the research are thus ambiguous, i.e. differences of allocation between indivisible and divisible objects (lines 82-85). Research aim is also unclear, what is the “improve allocation efficiency from the governor´s”? (lines 90-91). Actually, premises employed to substantiate the study are very vague. Practical implications should be outlined in the Introduction. Contributions in the field in comparison to the existing literature should be disclosed as well. It is very debatable the importance of the “ex-post verification” proposed by authors beyond an instrument to fine/reward water consumers.
- Subsection 3.1 should be located in the Introduction to enhance background. Assumption of lines 288-289 is not evident at all, it should be supported.
- Multiple editing mistakes were found. Variables must be defined at first appearance, Table 1 is confusing. Equations should be numbered and properly referred in the text. Visualization of Figure 1 should be improved. Check Figure 1. With the purpose of creating a degree of continuity with the editorial line of the journal, the number of references to articles published on Water should be increased.
A conceptual mathematical model was proposed to benchmark ex-ante against ex-post information related to water allocation. But the mechanism was not applied to any case study which diminishes the relevance of the article. Indeed, it is very unclear the contribution of the study in the field. Replicability and use by other researchers is therefore unlikely. Authors are suggested to reformulate the study by emphasizing its potential application instead of providing a large number of mathematical expression (some of them unnecessary).